# Connexin 43 Expression as Biomarker of Oral Squamous Cell Carcinoma and Its Association with Human Papillomavirus 16 and 18

**DOI:** 10.3390/ijms26031232

**Published:** 2025-01-30

**Authors:** Jose Roberto Gutierrez-Camacho, Lorena Avila-Carrasco, Idalia Garza-Veloz, Joel Monárrez-Espino, Maria Calixta Martinez-Vazquez, Roxana Araujo-Espino, Perla M. Trejo-Ortiz, Rosa B. Martinez-Flores, Reinaldo Gurrola-Carlos, Lorena Troncoso-Vazquez, Margarita L. Martinez-Fierro

**Affiliations:** Doctorate in Sciences with Orientation in Molecular Medicine, Academic Unit of Human Medicine and Health Sciences, Universidad Autónoma de Zacatecas, Zacatecas 98160, Mexico; rob_gutierrez_mm@uaz.edu.mx (J.R.G.-C.); idaliagv@uaz.edu.mx (I.G.-V.); jmonarrez@hotmail.com (J.M.-E.); calibgst@uaz.edu.mx (M.C.M.-V.); roxana.araujo@uaz.edu.mx (R.A.-E.); perlatrejo@uaz.edu.mx (P.M.T.-O.); rosabmartinez5@gmail.com (R.B.M.-F.); reinaldogc@gmail.com (R.G.-C.); loredoctoradomm@gmail.com (L.T.-V.)

**Keywords:** Connexin 43, oral squamous cell carcinoma, human papillomaviruses

## Abstract

Oral squamous cell carcinoma (OSCC) is the main form of head and neck cancer. Gap junctions (GJs) are communication channels involved in cell proliferation control; they consist of hemichannels formed by connexin (Cx) proteins. The abnormal expression/function of Cx43 has been associated with tumor progression. Also, some human papillomaviruses (HPVs) have been linked to squamous cell cancer. Therefore, this study aimed at assessing Cx43 as a potential OSCC biomarker and exploring its association with histopathological differentiation and HPV infection. OSCC samples were inspected using hematoxylin and eosin staining, and Cx43 expression and HPV 16/18 were tested by immunofluorescence. Pearson correlation tests, ANOVA, and Kaplan–Meier curves were used in the analysis. Samples from 39 patients with OSCC were studied. Most had well-differentiated histology and 61.5% were HPV+. Cx43 expression was significantly associated with HPV infection (*p* = 0.047), differentiation (*p* < 0.001), and survival (*p* = 0.009), and HPV positivity was also associated with the degree of differentiation (*p* = 0.012). Cx43 shows potential as a prognostic biomarker for OSCC. Lower Cx43 expression, correlated with poorer differentiation, is associated with an unfavorable prognosis. Further studies are needed to confirm its clinical utility.

## 1. Introduction

Oral squamous cell carcinoma (OSCC), which develops in the oral mucosa, is a common type of head and neck malignancy [1,2]. Compared with other tumors, existing therapeutic guidelines for head and neck cancers do not include molecular markers due to the lack of reliable predictive biomarkers. While there have been extensive efforts to find effective treatments, the need to identify clinically useful biomarkers still remains a research priority [3]. One area of current research is the role of cell junctions as these proteins are essential for cell function and communication, potentially leading to the identification of biomarkers and therapeutic targets. Connexins are proteins needed in the assembly of gap junctions (GJs) [4]. In humans, 21 isotypes of connexins (Cx) have been cloned with variations in size or charge selectivity [5]. Each Cx has four transmembrane domains, supplemented by two extracellular loops and a single cytoplasmic loop; both the N-terminal and C-terminal regions of the protein are situated within the cytoplasm [6].

The hemichannels, called connexons, are homomeric or heteromeric connexin hexamers positioned in the plasma membrane. The interaction of connexons between adjacent cells promotes the formation of GJ channels, facilitating gap junctional intercellular communication (GJIC) and enabling the transfer of small molecules including ions and second messengers [5]. GJIC is essential for the maintenance of tissue homeostasis, regulation of cell growth, and overall developmental processes [5].

While the structural composition of connexins defines their function in forming GJ, it also impacts their functional capability. In addition to their structural attributes, connexins have been shown to have a dual role, functioning as tumor suppressors while modulating intracellular signaling pathways, specifically Src [7]. Notably, a reduction in the expression levels of Connexin 43 (Cx43) has been associated with tumorigenesis and unfavorable outcomes in solid tumors, including in breast cancer [8,9]. The absence of GJIC can result in an increase in intracellular growth factors [10,11]. It has also been suggested that Cx43 facilitates cancer cell survival by mediating anti-apoptotic and growth signals, as well as promoting metastasis by modulating cancer cell motility and interactions with extracellular matrix components [12]. Cx43 may also control angiogenesis and immune cell evasion in the tumor microenvironment [13]. The GJ function of Cx43 has been shown to be involved in inhibiting proapoptotic signaling in cancer, where blocking Cx43 activity with RNAi has induced the apoptosis of hepatoma cells [14]. In addition, the pro-metastatic involvement of Cx43 GJIC has been described in several cancers [15,16,17]. Cx43 has been implicated in the regulation of intracellular pathways that modulate cell growth and cell death, in mechanistic signal transduction, and in the regulation of gene transcription [18]. For example, the C-terminal tail of Cx43 can interact with several oncogenes and may serve as a trigger for intracellular signaling cascades, control metabolic pathways, impact transcriptional control, and potentially be released into extracellular vesicles that exhibit paracrine cell–cell communication in the tumor microenvironment through its hemichannel function [19].

Despite it being susceptible to HPV infection, the oral cavity’s stratified epithelium demonstrates a lower tendency to undergo carcinogenic transformation resulting in OSCC [20]. The HPV family comprises circular, double-stranded DNA viruses with a genome of ~8000 base pairs that encode proteins critical to biological processes such as viral replication (E1 and E2/E4), virion assembly (L1 and L2), and several accessory proteins (E5, E6, and E7) [21]. High-risk HPV types, including HPV16, HPV18, HPV31, HPV33, HPV35, HPV39, HPV45, HPV51, HPV52, HPV56, HPV58, HPV59, and HPV68, have the ability to initiate carcinogenic transformation in the affected mucosal epithelium [21]. E6 and E7 expression is often linked to the integration of the viral genome into chromosomal regions characterized by genomic instability, leading to E2 coding region disruption and E6 and E7 dysregulation [22]. This mechanism allows HPVs to establish persistent infections and sustain replication as infected epithelial cells are terminally differentiated and thus do not produce viral particles. The phenomenon of nonproductive HPV infection plays a key role in the induction of tumorigenesis [22]. In contrast, low-risk HPV types are known to induce mucosal infections that are typically eradicated by the host’s immune system [23].

The suggestion that GJIC is impaired in HPV-related carcinogenesis was first mentioned by McNutt and Weinstein, who documented a decrease in the number of GJs in cervical carcinomas [24]. Subsequent research found alterations in Cx expression in HPV-mediated cervical intraepithelial neoplasia that also correlated with tumor severity [25]. The direct role of HPV in the reduced Cx in cervical tumor cells is based on the expression of high-risk HPV E6 and E5 oncoproteins. These oncoproteins alter the cellular distribution and phosphorylation pattern of Cx43, the most widely expressed Cx isoform [24,26]. Some studies have also linked the downregulation of Cx43 to impaired GJIC in cells expressing E6 and E5 [27], and Cx43 has then been suggested as indicative of epithelial dysplasia [28].

Building on the relationship between HPV oncoproteins and the altered expression and function of Cx43, this study aimed at assessing whether Cx43 can be used as a potential OSCC biomarker and at exploring its association with histopathological differentiation and HPV infection. While Cx43’s role in OSCC has been explored, the influence of HPVs on its subcellular localization remains unclear. Our findings provide new insights into this interaction, offering perspectives on OSCC progression mechanisms.

## 2. Results

The main characteristics of the study population are shown in Table 1. Out of the 39 cases with OSCC diagnosis, 19 (48.7%) were <65 years and 20 (53.3%) were ≥65 years (range 35–91 years) of old. There were 22 men (56.4%) and 17 women (43.6%). According to histological grading (i.e., degree of differentiation), seven (18%) were moderately differentiated, eight (20.5%) were poorly differentiated, and twenty-four (61.5%) were well differentiated. Nearly half of the patients (51.3%) were smokers and less than a third used alcohol (30.8%). Most tumors were located in the tongue (35.9%), followed by the floor of the mouth (20.5%). Most patients had a mild or absent inflammatory response (61.5%). Twenty-nine patients (74.4%) survived for less than 5 years. All the nine HPV+ samples were classified as well differentiated.

One-third of the tumors were classified as stage I (33.3%). An infiltrative invasion pattern was present in 66.7% of the tumors. Figure 1 illustrates the degrees of differentiation with Figure 1a,b showing a healthy mucosa, Figure 1c,d a well-differentiated OSCC, Figure 1e,f a moderately differentiated OSCC, and Figure 1g,h poorly differentiated OSCC.

A panel with images showing Cx43 protein staining according to the degree of differentiation and images of HPV-positive samples are shown in Figure 2. Cx43 immunofluorescence was performed in 39 OSCC cases and one healthy oral epithelium. The results were obtained by quantifying the fluorescence emitted by each image expressed in percentages. Cx43 was found in the cell membranes of healthy tissue showing 50% staining (Figure 2a) and in well-differentiated OSCC cases with an average staining of 54% (Figure 2e). In moderately differentiated OSCC, Cx43 expression decreased to 30% (Figure 2i), and in poorly differentiated OSCC staining, it decreased to 19% (Figure 2m). The expression of Cx43 throughout the epithelial cell layer was confirmed. Samples were also studied by immunofluorescence with monoclonal antibody C1P5 to detect the E6 protein of HPV16 and HPV18; nine carcinomas (23.1%) had positive nuclear and/or cytoplasmic staining (Figure 2q).

A correlation analysis including clinicopathological characteristics and Cx43 and HPVs is shown in Table 2. Relevant correlations between Cx43 expression and patient survival (*r* = 0.415; *p* = 0.009), HPV 16/18 (−0.32; *p* = 0.047), and degree of differentiation (*r* = 0.679; *p* < 0.001), were found, as was the correlation between HPV 16/18 and the degree of differentiation (*r* = 0.397; *p* < 0.012).

Kaplan–Meier plots with follow-ups after up to 15 years are shown in Figure 3. Survival according to the degree of differentiation of OSCC had a statistically significant difference with a value of *p* < 0.001. In addition, there was also a statistically significant difference in Cx43 expression with respect to survival with a value of *p* = 0.009.

The ANOVA compared mean survival years and the mean percentage of Cx43 expression (staining) by the three degrees of differentiation of OSCC (Figure 4). There was a trend by the degree of differentiation; higher differentiation was associated with higher survival and Cx43 expression means.

## 3. Discussion

This study assessed the potential of Cx43 as a biomarker for OSCC and its association with histopathological differentiation and HPV infection. Our epidemiological data showed that the age distribution of the patients with OSCC aligned with global trends although the male-to-female ratio did not reach the commonly reported 2:1 proportion [29]. Risk factors such as smoking and alcohol intake were measured, but not betel nut chewing, which is also a known risk factor for OSCC [30]. Alcohol acts as a cancer promoter, synergistically increasing the risk when combined with smoking. The combined effect of smoking, alcohol, and betel nut chewing raises the incidence of oral cancer by 123 times compared with abstainers [30]. In this study, substantial proportions of patients were smokers (51.3%) and alcohol consumers (30.8%); however, due to the absence of a control group, the association of these risk factors with OSCC could not be evaluated. In our cohort, the tongue was the most frequent OSCC site (35.9%), consistent with reports from the United States, Australia, Brazil, and Denmark [31,32,33], highlighting the need to promote clinical oral education and self-examination for early detection.

Our findings confirm that Cx43 localization and expression patterns in OSCC tissues differ from those in healthy oral mucosa. In non-dysplastic oral mucosa, Cx43 is predominantly localized at the cell membrane, decreasing with keratinization, potentially due to internalization and degradation processes [34]. In agreement with previous studies, OSCC samples in this study exhibited increased cytoplasmic Cx43 levels compared to controls, reflecting a shift from its typical membrane localization and loss of GJIC [35,36]. This loss may contribute to uncontrolled cell proliferation and tumor progression as Cx43 is known to function as a tumor suppressor in several cancers including head and neck squamous cell carcinoma (HNSCC) [37]. Cx43 expression was found to be positively regulated at early stages of carcinogenesis [36,38,39,40,41], with a decreasing trend observed as tumor differentiation worsened (well: 54%, moderate: 30%, poor: 19%), supporting the notion that lower Cx43 levels are associated with more aggressive OSCC forms. Unlike previous reports, 61.5% of tumors in this study were well differentiated [42,43], and 66.7% of patients exhibited an infiltrative pattern, consistent with prior findings [44]. Additionally, one-third of tumors were classified as stage I, which is associated with better prognosis and survival rates compared to advanced stages [45]. These findings align with previous reports demonstrating a significant reduction in Cx43 levels during carcinogenesis [46], though the underlying mechanisms remain unclear and could involve transcriptional, post-transcriptional, and degradation-related processes [47]. Importantly, our data confirm previous findings indicating a correlation between low Cx43 expression and poorer patient survival rates, reinforcing its potential prognostic value [3,8].

HPV16 and HPV18 are well-established independent risk factors for OSCC and other head and neck cancers [48,49,50]. Consistent with previous studies [51,52,53], our findings revealed a low prevalence of HPVs in OSCC (23.1%), which aligned with some reports but contrasted with others showing higher detection rates [54,55,56,57,58,59,60]. Interestingly, our results demonstrated a significant correlation between HPV infection and the degree of histological differentiation (*p* = 0.028), suggesting a possible role of HPVs in tumor biology. While HPV-positive OSCC cases have been associated with better prognosis and improved treatment responses [56,57,58], our study did not find a significant reduction in Cx43 levels in HPV-positive samples, likely due to the selection of samples based on histological differentiation.

Our findings related with Cx43 and HPV 16/18 suggest that HPV infection may alter Cx43 trafficking and localization in OSCC, potentially mediated by HPV oncoproteins such as E6. Previous studies have demonstrated that the E6 oncoprotein modulates Cx43 trafficking by binding to the human homologue of Drosophila discs (large) (hDlg), leading to impaired GJIC and altered protein localization [61,62,63,64,65,66,67]. These observations support the hypothesis that HPV-associated OSCC may exhibit distinct molecular characteristics compared to tumors driven by traditional risk factors such as smoking and alcohol consumption. Further research is needed to explore these mechanisms and their potential clinical implications.

The dysregulation of Cx43 occurs at multiple levels during cancer progression. At the transcriptional level, reduced expression can result from epigenetic silencing and transcription factor dysregulation [68]. mRNA stability and translation are influenced by cancer-associated microRNAs, and alternative translation initiation may produce truncated Cx43 isoforms with altered functions [19,69]. Post-translational modifications such as phosphorylation, ubiquitination, and SUMOylation further regulate Cx43 trafficking and stability at the plasma membrane [70,71]. Histopathological analysis in this study showed moderate-to-marked inflammation in 38.5% of cases, reflecting the dual role of innate immune cells in wound healing and tumor microenvironment modulation [72]. These regulatory complexities highlight the challenges of targeting Cx43 in cancer therapy.

Despite its tumor-suppressive role in early cancer stages, Cx43 can also promote malignant features and metastasis in advanced stages, raising concerns about its therapeutic modulation. Studies have explored the inhibition of Cx43 hemichannels using pharmacological agents such as Tonabersat and Meclofenamate, which show promise in attenuating brain metastases and are currently undergoing clinical trials [73,74]. Novel strategies targeting Cx43 hemichannels, such as carbon monoxide-based inhibition, represent a promising avenue for future research [74]. Low Cx43 expression has also been associated with poor prognosis and patient outcomes. In fact, low expression correlated with a wide range of established clinicopathological markers of poor prognosis, such as larger tumor sizes and poor differentiation, so it is believed that the most aggressive tumors had low or no Cx43 expression [75]. In clinical practice, our results could contribute to the determination of patient prognosis, but a latent challenge is to determine the role of various connexin isoforms in carcinogenesis and metastasis and to continue the search for new connexin modulators.

Finally, some limitations of this study should be recognized, and they included the lack of result validation using PCR, Western blot, or ELISA due to the availability of only formalin-fixed, paraffin-embedded samples. In the same way, due to the limited sample size, the findings presented in this study should be interpreted in this context. Future research with larger patient cohorts is necessary to validate these preliminary results and enhance the statistical power of the associations identified.

## 4. Materials and Methods

### 4.1. Study Design

This study included samples from patients who were histologically diagnosed with oral squamous cell carcinoma; all samples provided were verified to have complete clinicopathologic data and to be undamaged; in addition, a control sample of healthy tissue was obtained from the surrounding tissue of a tumor sample. The presence of HPV in the samples was not determinant for their inclusion in the study; its presence was identified until immunofluorescence was performed for its detection.

Formalin-fixed, paraffin-embedded tumor blocks (FFPEs) from biopsies of 39 patients with OSCC diagnosis were used. Tissue samples were obtained from pathology services at Zacatecas General Hospital and San Luis Potosí Central Hospital, Mexico. The data were extracted from the archives of the hospitals for which data were available before 2021. This retrospective study adhered to the Helsinki Declaration and was reviewed and approved by the Research Ethics Committee of Zacatecas Autonomous University (ID: CEI-UAMUyCS-01-2022) and the Research Committee of Zacatecas General Hospital (ID: 0033/2022).

### 4.2. Hematoxylin and Eosin Staining

Histological evaluation of FFPE tissues for preservation of morphology (general morphology and nuclear, cytoplasmic, and membrane details) was performed on 3 µm tissue sections using hematoxylin and eosin (H&E) staining and verified by at least three certified pathologists. Tissue sections were mounted on AUTOFROST (#20190710; Cancer Diagnostics Inc., Durham, NC, USA) loaded adhesion microscope slides.

FFPE tissue sections were deparaffinized at 65 °C for 30 min, followed by treatment with xylene for 20 min. Rehydration of tissue sections was performed with decreasing gradients of ethanol (100%, 90%, 70%, 70%, 50%, and distilled water). Tissue slides were stained by dipping in hematoxylin solution (#51275, Sigma-Aldrich, St. Louis, MO, USA) for 1 min, followed by washing with distilled water for 5 min. Counterstaining with eosin Y stain (#E4009, Sigma-Aldrich) was performed for 20 s. Dehydration with an increasing ethanol gradient was performed immediately after counterstaining. The slides were rinsed with xylene for 5 min and mounted with DPX universal mounting medium.

Images of histological samples stained with hematoxylin and eosin were captured using a bright field microscope (Carl Zeiss, Jena, Germany) at different magnifications (10× and 40×).

To determine the degree of differentiation of the tumors, the guidelines of the World Health Organization’s Classification of Tumors were used; these specify that squamous cell carcinomas are classified into three classes, well, moderately, and poorly differentiated. In well-differentiated squamous cell carcinoma, invasion is identified in the underlying tissue; it resembles normal epithelium and tends to form numerous keratin beads. Moderately differentiated squamous cell carcinoma shows marked nuclear pleomorphism, with mitotic activity including atypical mitoses. Poorly differentiated squamous cell carcinoma shows the predominance of immature cells with numerous typical and atypical mitoses; keratinization is minimal [76].

To determine the inflammatory response, it was necessary to identify the presence of cellular infiltrate in acute course (neutrophils) and chronic course (lymphocytes, plasma cells, or macrophages), in addition to the distribution (focal, multifocal, diffuse, or locally extensive) and cellular necrosis [77].

### 4.3. Detection of Cx43 by Immunofluorescence

Paraffin-embedded tumor tissues were serially sectioned at 3 μm. For each FFPF, a histological slice selected by the pathologists was evaluated and its precise degree of differentiation was identified for the identification of Connexin 43. FFPE tissue sections were deparaffinized at 65 °C for 30 min, followed by treatment with xylene for 20 min, and rehydrated in an ethanol series. Antigen unmasking was performed in a 0.1 M Tris 0.01 M EDTA buffer (pH 9.0) using an electric pressure cooker (Avair, Biofa, Veszprem, Hungary) for 10 min at ~105 °C followed by 10 s digestion in 0.25% Gibco trypsin phenol red (1:50; Life Technologies, Carlsbad, CA, USA, Ref: 25050–014). After a protein blocking step for 20 min, the slides were incubated overnight at 4 °C using Cx43 antibodies (1:100, #3512, Cell Signaling, Beverly, MA, USA). The fluorochrome-labelled secondary antibody, goat anti-rabbit IgG Alexa Fluor 546 (red) (A11035), was applied at 1:200 for 60 min.

Subsequently, the slides with tumor tissues were incubated overnight at 4 °C with mouse monoclonal HPV16/18 E6 (C1P5) primary antibody (Santa Cruz Inc., Santa Cruz, CA, USA) in blocking buffer at a dilution of 1:100. After extensive washing, sections were incubated with goat anti-mouse secondary antibody conjugated to Alexa 488 (Invitrogen, Carlsbad, CA, USA) for 60 min at 1:200 dilution. Slides were washed and mounted in Vectashield medium for fluorescence (Vector Labs. Inc., Newark, CA, USA). All incubations were performed in a humid chamber.

The integrated density value (% staining) of the Cx43 immunostaining for FFPE-fixed OSCC tissues was calculated using the color deconvolution tool ImageJ (NIH, Bethesda, MD, USA) (H DAB filter); for HPV detection, samples were classified as positive or negative according to a cut-off point of detection of intense nuclear and/or cytoplasmic staining in tumor cells. The pipeline used to process the images and determine the percentage of staining was to convert the images to 8 bits, generate a binary mask, and then select a filter by selecting the options for process, filter, and Gaussian blur; then, the image was inverted in the edit option; then, the image, adjustment, and threshold were selected; finally, the software was selected to analyze and measure the data. The software yielded, in an excel table, the percentage of staining.

The slices were evaluated via images captured with an Axiocam 503 color camera attached to an Axio Observer Z1 motorized inverted fluorescence microscope with an LD Plan-NEOFLUAR 20x/0.4 Ph2 Korr objective. The performed analyses were conducted by using ZEN Pro software (version 3.11, All Carl Zeiss, Jena, Germany). It was decided to perform the immunofluorescence test for the detection of HPV 16/18 and Cx43 in order to visually contrast their locations in the cell.

### 4.4. Statistical Analysis

Descriptive statistics of patients’ general characteristics were expressed as frequencies and percentages. Pearson’s correlation tests were used to test correlations between clinicopathological variables and Cx43 and HPV fluorescence intensities. For survival analysis, Kaplan–Meier estimations were used. ANOVA tests were also performed to identify mean differences of the clinicopathological characteristics and degrees of differentiation of OSCC. Statistical analyses were performed using IBM SPSS Statistics version 13.0 (SPSS Inc., Chicago, IL, USA) and SigmaPlot v12.0 Software (Systat Software Inc., San Jose, CA, USA). All *p*-values < 0.05 were considered statistically significant.

## 5. Conclusions

Cx43 could play a prognostic role in OSCC, particularly in relation to its level of differentiation, whereby well-differentiated carcinomas could have more favorable prognosis and longer survival compared with poorly differentiated carcinomas. The findings were also consistent with data showing that HPV16/18 E6 may influence the trafficking of Cx43 to the plasma membrane, where it becomes trapped into the cytoplasm. Overall, it is possible that HPV16/18 E6 expression regulates Cx43 trafficking, leading to the reduced supply of hemichannels to the plasma membrane and inhibiting junction formation, therefore promoting tumor progression. Yet, further research is needed to confirm the potential role of Cx43 as a predictive marker.

## Figures and Tables

**Figure 1 ijms-26-01232-f001:**
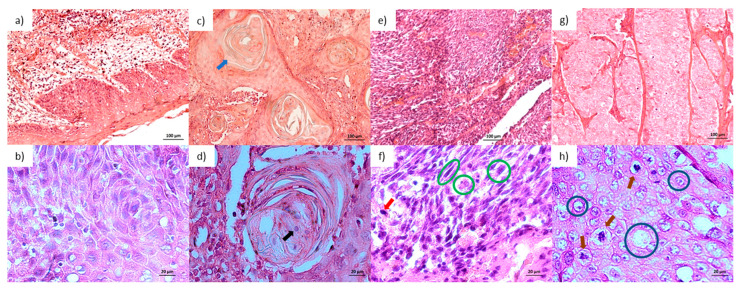
Degrees of differentiation of oral squamous cell carcinoma: (**a**,**b**) healthy oral mucosa; (**c**,**d**) well-differentiated oral squamous cell carcinoma (OSCC) with keratinization (blue arrow), nuclear hyperchromatism (black arrow), mild pleomorphism, and few mitoses; (**e**,**f**) moderately differentiated OSCC with moderate pleomorphism (green circles) and abundant typical and atypical mitoses (red arrow); and (**g**,**h**) poorly differentiated OSCC with marked pleomorphism (blue circles) and typical mitoses (brown arrows).

**Figure 2 ijms-26-01232-f002:**
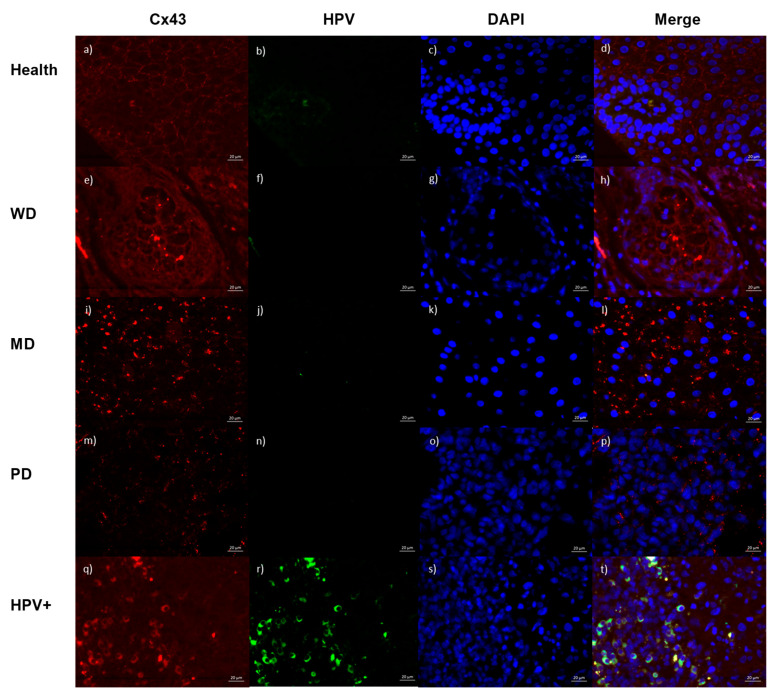
Immunofluorescence for Cx43 (Connexin 43) (Alexa 546, red) and for HPV16/18 (Human Papillomavirus 16/18) E6 protein (Alexa 488, green) in oral squamous cell carcinoma: (**a**–**d**) healthy oral mucosa; (**a**) Cx43 localized predominantly in cytoplasmic membrane with granular appearance pattern; (**b**) negative HPV staining, with a small amount of keratin-containing stain identified; (**c**) DAPI (4′,6-diamidino-2-phenylindole) (blue) showing nuclei of homogeneous shape and size; (**d**) during merge, nuclei were observed surrounded by cytoplasmic membrane demarcated by Cx43; (**e**–**h**) well-differentiated carcinoma; (**e**) Cx43 showing similar distribution of cytoplasmic membrane as healthy tissue, but granular pattern is coarser, and a small expression is also seen in cytoplasm; (**f**) negative HPV staining; (**g**) DAPI showing nuclei with variation in size and shape with irregular distribution; (**h**) merge showing predominance of connexin in membrane with less expression in cytoplasm; (**i**–**l**) moderately differentiated carcinoma; (**i**) Cx43 showing irregular expression in cytoplasm and membrane with course granular pattern; (**j**) negative HPV expression; (**k**) DAPI showing nuclei with variation in size and shape; (**l**) merge showing irregular distribution of Cx43 with respect to nucleus, with irregular cytoplasmic and membranal localization; (**m**–**p**) poorly differentiated carcinoma; (**m**) Cx43 showing decreased expression in both membrane and cytoplasm with respect to moderately differentiated carcinoma, with a fine granular pattern; (**n**) negative HPV expression; (**o**) DAPI showing increased cell density, including nuclei with marked irregularity in both shape and size as well as in dye affinity; and (**p**) merge showing that some cells have completely lost Cx43 expression while there is a low quantity in the other cells. (**q**–**t**) HPV-infected OSCC. (**q**) Cx43 shows a predominantly cytoplasmic distribution without the granular pattern seen in non-HPV-infected carcinoma and healthy tissue. (**r**) A cytoplasmic distribution of HPV infection similar to the distribution expressed by connexin is observed. (**s**) DAPI shows that high cell density, size, and density of nuclei are not uniform. (**t**) Merge shows that the cytoplasmic distribution of Cx43 and HPV expression are similar in terms of location.

**Figure 3 ijms-26-01232-f003:**
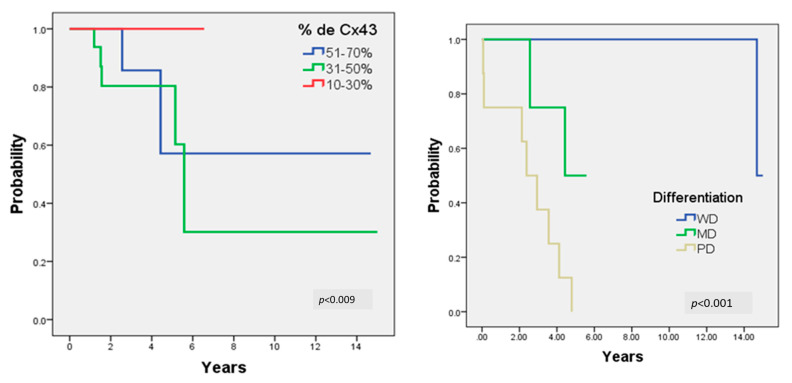
Kaplan–Meier survival plots: (**left**) increasing percentage of Cx43 expression shows higher survival (*p* = 0.009); (**right**) patients with well differentiation (WD) have higher survival (*p* < 0.001) compared with moderately (MD) and poorly differentiated (PD) tumors.

**Figure 4 ijms-26-01232-f004:**
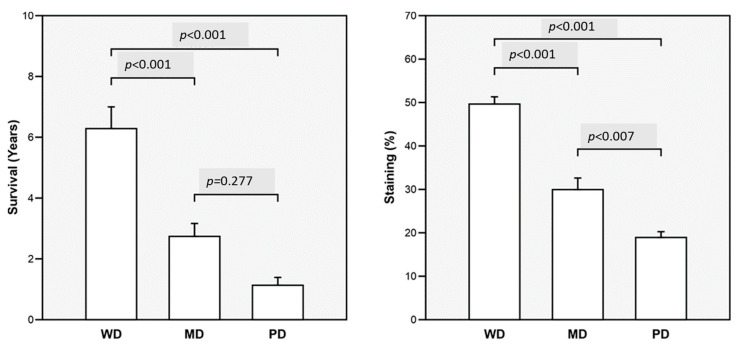
Comparison of survival years and Cx43 expression by differentiation stages of OSCC; (**left**) mean survival years by degree of tumor differentiation; (**right**) mean percentage expression of Cx43. WD: well differentiated; MD: moderately differentiated; PD: poorly differentiated.

**Table 1 ijms-26-01232-t001:** Main clinicopathological characteristics of OSCC patients.

Variable Category	Frequency (%), *n* = 39
Age in years <65 ≥65	19 (48.7) 20 (53.3)
Sex Male Female	22 (56.4) 17 (43.6)
Tumor Location Palate Tongue Lower lip Gum Floor of mouth	7 (17.9) 14 (35.9) 5 (12.8) 5 (12.8) 8 (20.5)
Smoking Yes No	20 (51.3) 19 (48.7)
Alcohol intake Yes No	12 (30.8) 27 (69.2)
TNM ^1^ staging I II III IV	13 (33.3) 7 (17.9) 10 (25.6) 9 (23.1)
Invasion pattern Infiltrative Cohesive	26 (66.7) 13 (33.3)
Inflammatory response Absent/mild Moderate/marked	24 (61.5) 15 (38.5)
Survival in years ≥5 <5	10 (25.6) 29 (74.4)
Connexin 43 in % 10–30 31–50 51–70	12 (30.8) 15 (38.5) 12 (30.8)
Human Papillomavirus 16/18 Positive Negative	9 (23.1) 30 (76.9)
Degree of differentiation Well Moderate Poor	24 (61.5) 7 (18.0) 8 (20.5)

^1^ TNM: Tumor, lymph node, and metastasis staging system.

**Table 2 ijms-26-01232-t002:** Correlation between clinicopathological characteristics of patients with OSCC and Cx43 and human papillomavirus (HPV).

Variables	Connexin 43		HPV 16/18	
	*r* (95%CI)	*p*-Value	*r* (95%CI)	*p*-Value
Age	0.197 (−0.110–0.509)	0.229	−0.022 (−0.380–0.317)	0.896
Sex	−0.049 (−0.407–0.270)	0.769	0.113 (−0.207–0.399)	0.492
Tumor location	−0.227 (−0.520–0.069)	0.164	0.103 (−0.254–0.415)	0.534
Smoking	0.162 (−0.144–0.471)	0.326	−0.197 (−0.462–0.118)	0.230
Alcohol intake	0.123 (−0.218–0.424)	0.456	−0.233 (−0.443–0.051)	0.153
Tumor staging	−0.091 (−0.389–0.242)	0.581	0.232 (−0.065–0.521)	0.155
Invasion pattern	0.066 (−0.269–0.373)	0.689	0.129 (−0.192–0.370)	0.433
Inflammatory response	−0.009 (−0.346–0.311)	0.956	−0.183 (−0.445–0.154)	0.265
Survival	**0.415 (0.127–0.700)**	**0.009**	−0.354 (−0.706–−0.085)	0.559
HPV 16/18	**−0.32 (−0.519–−0.087)**	**0.047**	-	-
Degree of differentiation	**0.679 (0.556–0.790)**	**<0.001**	**−0.397 (−0.542–−0.254)**	**0.012**
Connexin 43	-	-	**−0.32 (−0.519–−0.087)**	**0.047**

Statistically significant Pearson correlation coefficients are indicated in bold.

## Data Availability

The data that support the findings of this study are available on request from the corresponding authors (M.L.M.-F. and L.A.-C.).

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
