# Peer review of "Connexin 43 Expression as Biomarker of Oral Squamous Cell Carcinoma and Its Association with Human Papillomavirus 16 and 18"

_ijms, 2025, doi:10.3390/ijms26031232_

Round 1
Reviewer 1 Report
Comments and Suggestions for Authors
The manuscript investigates the possibility of using Connexin 43 (Cx43) as a biomarker of oral squamous cell carcinoma (OSCC) and the correlation with human papilloma infection. The role played by Cx43 in the development and progression of different types of cancer has been the subject of several studies, the reported results suggest a double function, as a suppressor or pro-tumorigenic factor mainly in relation to the stage of cancer differentiation. Also with regard to OSCC, the involvement of Cx43 has been the subject of in-depth studies (10.3892/ijo.2014.2394; doi: 10.3390/cancers15204924). For this reason, the results on the expression of Cx43 are not new in the field, and those on the correlation with Cx43 and HPV are the most interesting. In any case, the fact that the number of samples is small makes the results very preliminary. The analysis should focus on the aspect that HPV infection can influence the subcellular localization of Cx43 mainly in relation to the different degree of differentiation/staging of cancer. Minor critical points. 1) The introduction and discussion are too long and should be more focused on the research topic. 2) In some cases, the references are not recent. In particular, the different results obtained regarding the expression of Cx43 in various stages of cancer compared to previous reports in the literature should be discussed. 3) Since Cx43 expression decreases during OSCC progression, it is difficult to talk about Cx43 as a biomarker of OSCC. 4) It is wrong to talk about "cytoplasmic expression" and "membrane expression". The expression has been replaced by "localization". 5) The manuscript should be carefully revised (OSSC instead of OSCC).
Author Response
We have attached the answer as a word file

Reviewer 2 Report
Comments and Suggestions for Authors
The reviewer would like to acknowledge the work conducted by the authors, highlighting that the subject is extremely relevant, since biomarkers are instrumental in cancer diagnosis, treatment and prognosis. Nonetheless, there are some questions that need to be addressed prior to consider its publication.
1. The major concern of the reviewer relates to quantification of the connexin 43.
1.1 Connexin 43 expression was determined within histological sections via immunofluorescence technique. Material and methods section do not provides details on the following: how many histological section per biopsy were evaluated. This should be corrected accordingly.
1.2 Authors state that Image J deconvolution tool was used to determine the value of integrated density. It is not entirely clear how this was achieved. Authors must provide the pipeline used in Image J, in order to assess the validity of the attained data.
1.3 Since integrated density was determined using Image J, it is not clear for what purpose ZEN Pro software was used. Details in this regard should be added.
1.4. Authors have based the quantification of connexin 43 expression solely relying in image-based methods. No validation using molecular biology methods (western blot, PCR,..) is provided. Although it is not fatal, it must fully acknowledged in the discussion/conclusion as a study limitation, highlighting the need for validation of the attained data.
2. It is not clear how the degree of differentiation of the tumors was determined. This should be corrected accordingly.
3. Table 1. Terms as "well" and "poor" would be more accurate if "high" and "low"
4. Table 1. It is not clear how the variable "inflammatory response " (which is discussed in the discussion section) was determined. Details should be added.
Author Response

(The authors gave the same response as above.)

Reviewer 3 Report
Comments and Suggestions for Authors
Manuscript entitled "Connexin 43 Expression as Biomarker of Oral Squamous Cell Carcinoma and Its Association with Human Papillomavirus 16 and 18" by José Roberto Gutiérrez-Camacho et al.
This manuscript investigates Connexin 43 (Cx43) expression as a prognostic biomarker in oral squamous cell carcinoma (OSCC) and its correlation with histopathological differentiation and HPV 16/18 infection. The study addresses an important topic in cancer research, but several areas require further elaboration for clarity, depth, and impact.
Comments:
1. In introduction expand on the molecular mechanisms by which Cx43 influences tumor progression beyond its gap junctional intercellular communication (GJIC) functions.
2. Clarify the criteria for patient inclusion, particularly regarding HPV status determination.
3. Discuss whether other factors, such as comorbidities or treatment regimens, could influence the observed relationships.
4. Discuss the potential mechanistic interactions between HPV oncoproteins (e.g., E6, E7) and Cx43, particularly regarding intracellular localization and degradation pathways.
5. Address the clinical implications of reduced Cx43 expression in poorly differentiated OSCC, particularly for early detection and targeted therapy.
6. The manuscript lacks explicit discussion of knowledge gaps and future research directions.
7. Expand on how these findings could influence clinical practice.
8. Develop a summary figure.
9. Incorporate subgroup analyses to assess whether HPV+ tumors exhibit distinct patterns of Cx43 expression across differentiation grades
10. Add a scale bar in Figure 1.
11. In Table 2 add confidence intervals and effect sizes
Author Response

(The authors gave the same response as above.)

Round 2
Reviewer 2 Report
Comments and Suggestions for Authors
The reviewer considers that the raised concerns were properly addressed by the authors.
Comments on the Quality of English LanguageThe reviewer advises revision in this regard.
Reviewer 3 Report
Comments and Suggestions for Authors
The authors have adequately addressed my comments, and the manuscript can be accepted for publication.